biomaterials/materials science

coupled, effect, particle, source, materials, calcinations

**Author for correspondence:**
Samina Ahmed
e-mail: shanta_samina@yahoo.com

This article has been edited by the Royal Society of Chemistry, including the commissioning, peer review process and editorial aspects up to the point of acceptance.

# Coupled effect of particle size of the source materials and calcination temperature on the direct synthesis of hydroxyapatite

Md. Sahadat Hossain[1], Monika Mahmud[1],
Sazia Sultana[1], Mashrafi Bin Mobarak[1], M. Saiful Islam[2]
and Samina Ahmed[1,2]

[1]Institute of Glass and Ceramic Research and Testing (IGCRT), and [2]BCSIR Laboratories, Dhaka, Bangladesh Council of Scientific and Industrial Research (BCSIR), Dr. Qudrat-i-Khuda Road, Dhanmondi, Dhaka-1205, Bangladesh

MSH, 0000-0001-8273-8559; MM, 0000-0002-1683-4320;
SS, 0000-0003-4269-2554; MB, 0000-0002-9488-9802;
MSI, 0000-0003-4913-5937; SA, 0000-0001-6626-3610

We report the effect of controlled particle size (obtained by using 80, 100, 120, 140 and 200 mesh) of the source materials on the synthesis of a well-known biomaterial, hydroxyapatite (Hap). In addition to this, we have also mapped the consequence of applied temperature (700°C, 800°C and 900°C) on the crystallographic properties and phase composition of the obtained Hap. Nevertheless, although with Hap, in each case, β-tricalcium phosphate (β-TCP) was registered as the secondary phase the ANOVA test revealed that the results of the crystallographic parameters are significantly different for the applied sintering temperature 700°C and 800°C ($p < 0.05$), while the data obtained for calcination temperature 800°C are not significantly different from that acquired at 900°C ($p > 0.05$). Fourier transform infrared spectrophotometer data ensured that, irrespective of mesh size and calcination temperature, the synthesized Hap samples were of carbonated apatite with B-type substitution. Interestingly, for all cases, the % of carbonate content was below the maximum limit (8%) of the $CO_3^{2-}$ ion present in bone tissue hydroxyapatite.

## 1. Introduction

Being similar to the mineral composition of bone, hydroxyapatite (Hap) in recent times has found multi-dimensional applications

covering a broad zone from biomedical to environmental fields. The aspiration to use Hap for the treatment of damaged organs or bones has prompted researchers to expand research on this biomaterial [1–4]. Owing to its very special properties like biocompatibility, bioactivity, osteo-conductivity, nontoxicity, bone healing function etc., Hap is being used in the field of biomedical research [5–13]. It has also found other notable applications in chromatography, biosensors, gas sensor, catalysts, fuel cells, adsorbents, etc. [14]. In particular, it has been widely used for the removal of heavy metals [15,16] and arsenic from wastewater [17]. Regarding the importance of Hap, day by day researchers are exploring their focus on developing various synthetic routes using either chemical sources or biogenic resources/ bio-wastes [10,18–20].

Among the developed methods, chemical precipitation, solid-state thermal, sol-gel, hydrothermal, micro-emulsion and microwave irradiation methods have been categorized as top-ranking methods for Hap synthesis. Nevertheless, coupled with the synthetic routes, the selection of raw materials is also a vital issue to consider. A number of previous studies used various Ca-salts and phosphate-salts or phosphoric acid as Ca and P precursor, respectively. However, at present, researchers prefer using Ca-enriched biogenic resources (e.g. eggshell, bovine bone, fish bone, coral shell, etc.) for synthesizing Hap. Indeed, due to the high demand of Hap, the awareness of using waste materials to synthesize this biomaterial is increasing day by day. Obviously, such diversification of raw materials and synthesis conditions as well as methods causes the structural properties of Hap to differ slightly. To the best of our knowledge, the raw materials (either biogenic or chemical) used so far to synthesize Hap have never been subjected to sieve analysis.

In this present work, we focused on using starting materials of the same particle size to facilitate a better fusion process during sintering. Our intention was to examine whether such an approach affects the synthesis procedure of Hap by minimizing the calcination temperature.

# 2. Material and methods

## 2.1. Materials

Ca-precursor (eggshell) was collected from a local restaurant while phosphorus source, i.e $(NH_4)_2HPO_4$, was purchased from E-Merck Germany. The latter source was analytical grade and used as received. Conversely, before using the eggshell (ES), it was cleaned with plenty of water and ovendried at 105°C then milled to a fine powder. Any required amount of deionized water was prepared in the laboratory via a double distillation process.

## 2.2. Processing of the raw materials and synthesis of hydroxyapatite

Upholding the Ca/P ratio at 1.67, the requisite amount of ES powder and $(NH_4)_2HPO_4$ was first mixed manually and then the dry mixture was ball milled for 4 h at 550 rpm (Model: Pulverisette 5 classic line Planetary Ball Mill). Subsequently, sieve analysis was done by sifting the ball-milled mixture through a stack of wire mesh sieves. The size of the sieves used to get distinct sized particles ranged from 80 to 200 mesh. A sieve shaker facilitated the vibration of the sieve stack for a precise time and thus permitted the unevenly shaped particles to be orientated accordingly while passing through the sieves. Furthermore, the distress of the sieves helped to break apart any agglomeration and, consequently, allowed a more consistent measurement of the particle size distribution. Extra care was taken in selecting a suitable agitation period, so that no particle fracture occurred. Each set of sieved particles were then divided into three portions and individually subjected to calcination operation at three different temperatures, i.e. 700°C, 800°C and 900°C. Such solid-state calcinations resulted in the formation of desired Hap which was ensured through several characterization techniques, as described in the following sections, while the entire experimental approach is depicted in figure 1.

## 2.3. Characterization of eggshell powder and synthesized hydroxyapatite

Using various techniques, e.g. wavelength-dispersive x-ray fluorescence (WDXRF), x-ray diffractometer (XRD), Fourier transform infrared spectrophotometer (FT-IR), scanning electron microscopy (SEM) and thermo-gravimetric (TG) analyses, characterizations of ES and Hap were accomplished.

To acquire the chemical composition of ES, WDXRF (ZSX Primus IV, Rigaku) was operated at a tube current of 100 mA coupled with 50 kV. The diffraction pattern was used to measure the respective

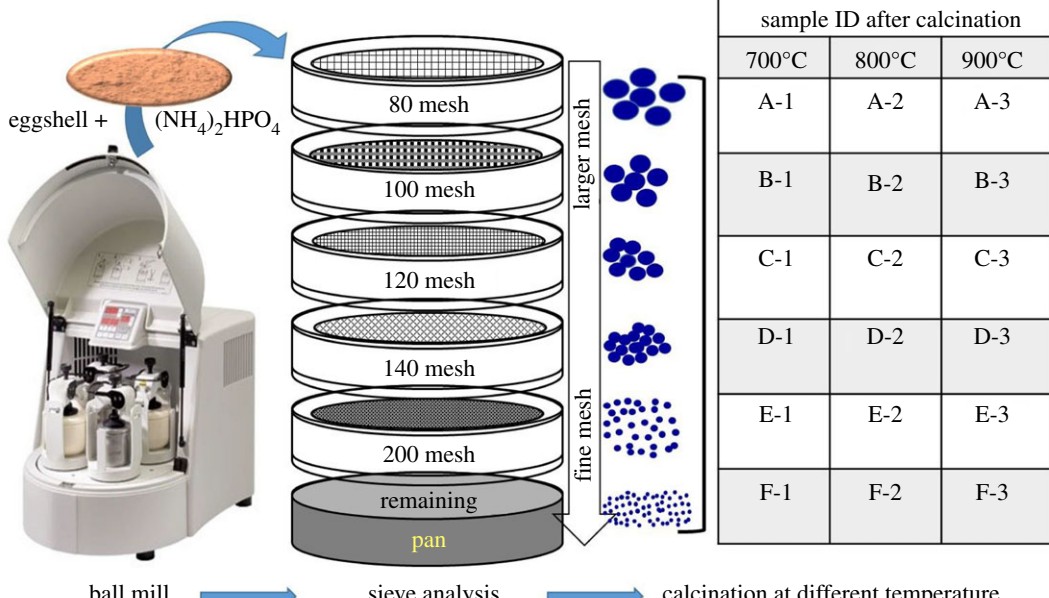

**Figure 1.** Experimental set-up to synthesize Hap.

chemical constituents in oxide form. The samples (raw ES and ES calcined at 900°C for an hour) were ground into a fine powder and compressed into the sample mould to fabricate the pellets for analysis.

The branding of corresponding phases of ES and Hap were certified by the XRD (PANalytical X'Pert PRO XRD PW 3040). Cu K$\alpha$ radiation ($\lambda$ = 1.5406 A) with a step scan of 4°/min enabled the collection of intensity data within the chosen scanning range, $2\theta = 5°–75°$. Recorded data were validated by comparing with standard JCPDS files. Earlier methodologies [10] were followed to record the FT-IR band positions and Raman shifts. An FT-IR Prestige 21 (SHIMADZU) equipped with attenuated total reflection (ATR) set-up and Raman spectrometer (HORIBA MacroRAM™ Raman Spectrometer) was used. The TG analysis was carried out with Perkin Elmer Pyris 1 TGA. Fixing the heating rate at 20°C min$^{-1}$, the TG profile was logged from 30°C to 950°C under a nitrogen atmosphere. The surface morphology together with microstructural arrangements was captured by SEM (Phenom Pro) setting the accelerating voltages at 5 kV, 10 kV and 15 kV.

## 2.4. Statistical analysis

Using ANOVA, single-factor test statistical analysis was executed to evaluate the existence of any significant difference among the crystallite size, dislocation density, % of Hap and β-TCP, volume fraction of β-TCP, degree of crystallinity, crystallinity index and micro-strain of all the Hap samples. A value of $p < 0.05$ was taken into account as statistically significant, while $p > 0.05$ was regarded as statistically insignificant.

## 3. Results and discussion

### 3.1. Characterization of source materials

Given in figure 2$a$,$b$ are the XRD patterns of ES powder: (i) ovendried at 105°C for 5 h and (ii) calcined at 900°C with an increment of temperature, 3°C min$^{-1}$ for 1 h. The diffractogram as displayed in figure 2$a$ represents well the presence of trigonal calcite (JCPDS No. 00–047-1743) as the principal phase [21–23], and no other crystalline phase was detected. On the contrary, upon calcination at 900°C, this calcite phase transformed into CaO by eliminating $CO_2$ (equation (3.1)) which is clearly visualized (figure 2$b$) from the observed $2\theta$ position at 37.35° (JCPDS No. 00-037-1497) [21]. Interestingly, although ES additionally contains some other oxides/carbonates/phosphates of Mg, Na, Zn etc. in insignificant ratio as detected by WDXRF (inset Tables of figure 2$a$,$b$), the presence of these components were not

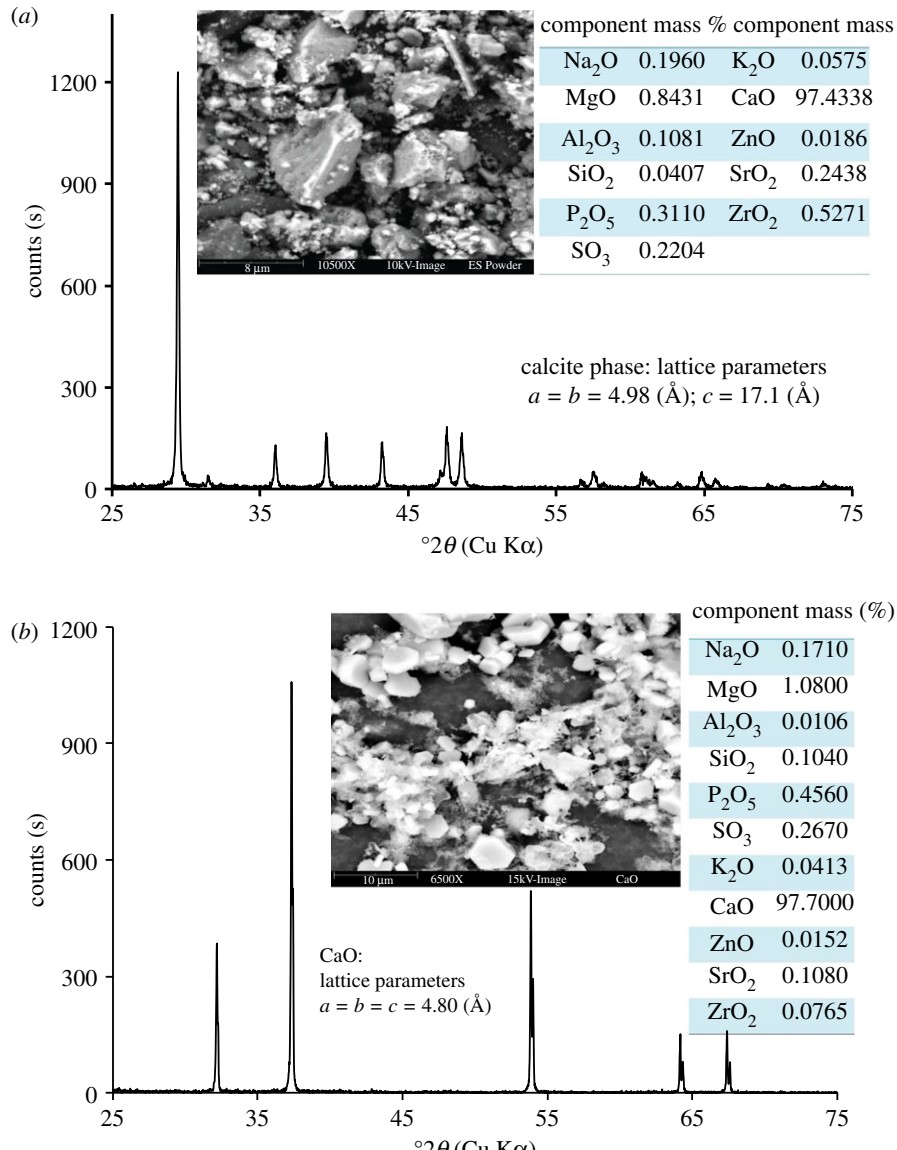

**Figure 2.** XRD pattern of ES; (*a*) ovendried; (*b*) calcined.

envisaged in the XRD patterns. Perhaps the detection limit of these constituents is further down the range of powder XRD.

$$CaCO_3 \rightarrow CaO + CO_2 \uparrow. \tag{3.1}$$

The microstructural information of ovendried and calcined ES was examined to achieve the morphology of its constituents. The inset images as depicted in figure 2*a*,*b* represent the SEM micrograph of the calcite and CaO phases, respectively. In the case of calcite (figure 2*a*), the particles were observed as aggregated smaller fractions tied together but for calcined ES (figure 2*b*), the particles were mostly geometrically shaped. The presence of organic substances (e.g. glycoproteins) inside the calcified layers along with some imprisoned water subsisting at the grain boundary of ovendried ES are possibly responsible for agglomerating the particles, while upon thermal treatment elimination of organic components occurred and CaO particles appeared with their geometrical shape.

Following these observations, the next ES powder was subjected to thermal analysis and the resultant TG curve signifying the weight losses at different intervals of selected temperature ranges is shown in figure 3.

Starting from the initial temperature (30°C) to 195°C, a very slow decrease in mass % (1.65%) is visualized in the TG profile which is attributed to the dehydration of the sample [21,24,25]. The weight loss for a second time continued up to 460°C with the similar relaxation mode and

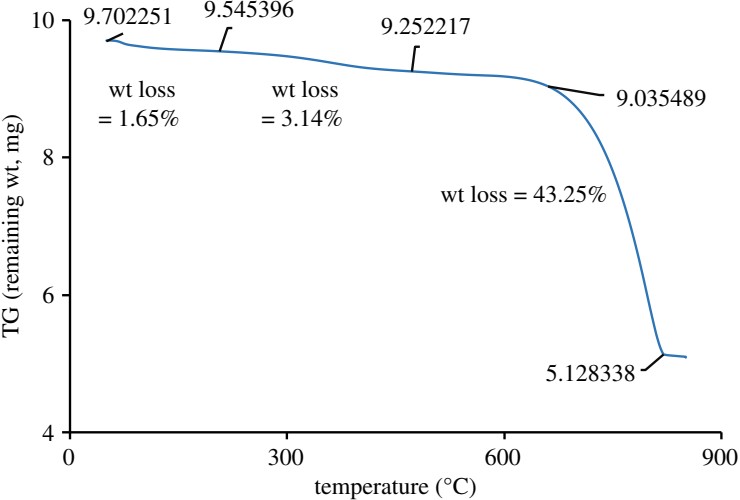

**Figure 3.** TG characteristics of ES.

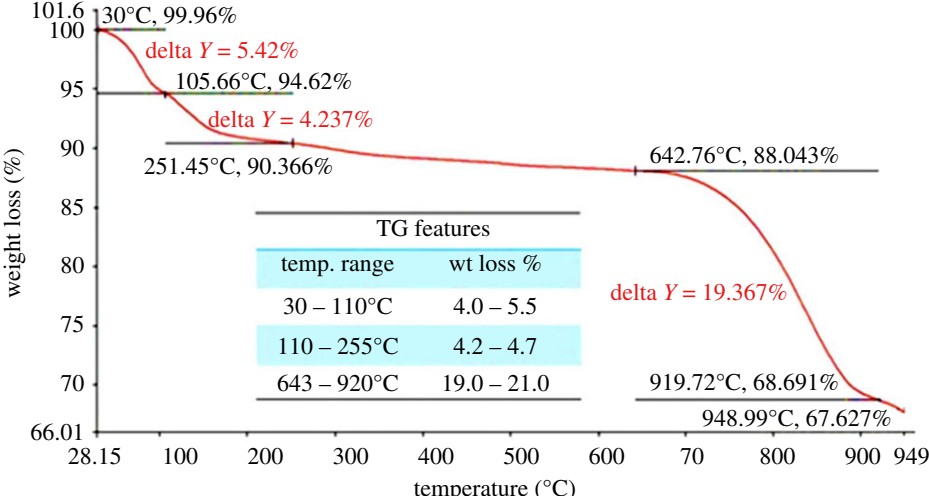

**Figure 4.** TG characteristics of ES and $(NH_4)_2HPO_4$ mixture.

demonstrated the decomposition of organic protein materials. Nevertheless, since decomposition of $CaCO_3$ above 600°C let $CO_2$ become free according to equation (3.1), a substantial weight loss (43.25%) occurred between 650° and 850°C, and this observation was in tune with previous research [25]. Using this mass loss associated with the discharge of $CO_2$, the percentage of $CaCO_3$ (98.28%) present in the ES was calculated with the aid of equation (3.2) [24].

$$\% \ CaCO_3 \ = \% \ x \ CO_2 \ \frac{W_{CaCO_3}}{W_{CO_2}}, \qquad (3.2)$$

where $\%x \ CO_2$ is the mass loss due to the discharge of $CO_2$ and $W_{CaCO_3}$ and $W_{CO_2}$ are the molar mass (in g/mol) of $CaCO_3$ and $CO_2$, respectively.

Here in this research work, we intended to examine the coupled effect of particle size of the source materials and calcination temperature to synthesize Hap by the solid-state method. Hence, it was necessary to accomplish the thermal analysis of ES and $(NH_4)_2HPO_4$ mixture of various mesh sizes of interest which enabled us to pinpoint the respective temperature in each case at which the chemical reaction for Hap formation takes place. A typical TG graph of the mixture containing ES and $(NH_4)_2HPO_4$ sieved with 200 mesh is displayed in figure 4, while the inset table summarizes the respective TG features of all samples. The associated mass loss behaviours of all TG profiles (inset table) were of a similar trend particularly confined with three distinctive weight losses in three specific regions. The first weight loss (4–5.5%) as noticed within ambient temperature to 110°C was due to the elimination of water of crystallization, trapped moisture and ammonia which continued to decrease

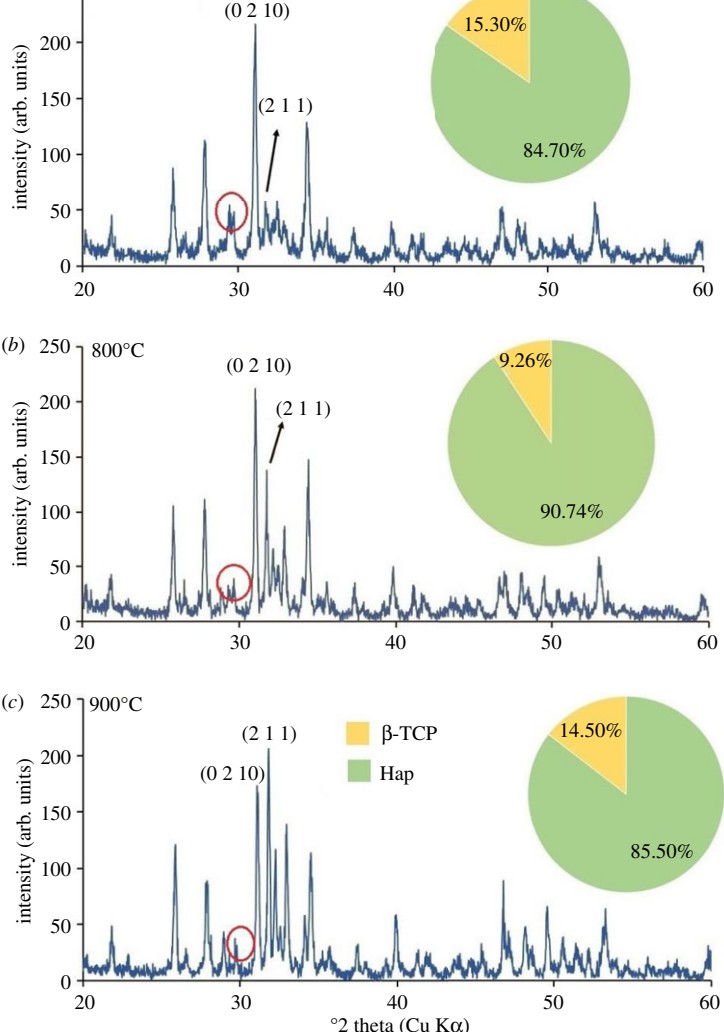

**Figure 5.** XRD pattern of Hap samples; (*a*) A-1; (*b*) A-2; (*c*) A-3.

with an additional percentage (4.2–4.7%) up to 251°C [26,27]. A mass loss of about 2.0% appeared with a steady fashion in the temperature range 252–632°C is ascertained for the decomposition of $HPO_4^{2-}$ according to reaction (3) [27]. Finally, the most prominent weight loss was recorded in the temperature scale 643–920°C. Obviously, this weight loss is the symptom of apatite formation reaction. At this stage, ES ($CaCO_3$) starts to be decarboxylated [26] releasing $CO_2$ allowing equation (3.1) to proceed. Furthermore, the weight loss was also supplemented either by the removal of interstitial water or owing to the breakdown of $P_2O_7^{4-}$ which ultimately takes part in the reaction to form Hap [27] according to reaction (4). Leaving about 67–68% of the residual mass, around 20% mass loss was observed at this stage.

$$HPO_4^{2-} \rightarrow P_2O_7^{4-} + H_2O \tag{3.3}$$

and

$$CaO + P_2O_7^{4-} + H_2O = Ca_{10}(PO_4)_6(OH)_2. \tag{3.4}$$

## 3.2. Characterization of synthesized hydroxyapatites

### 3.2.1. X-ray diffractometer analysis: phase identification and structural illustration

XRD patterns of all Hap samples prepared using 80–200 mesh sized raw materials calcined at three different temperatures (700°, 800° and 900°C) are shown in figures 5–10. The diffractograms were in line with the usual impression of temperature effect, i.e. as the sintering temperature headed for the

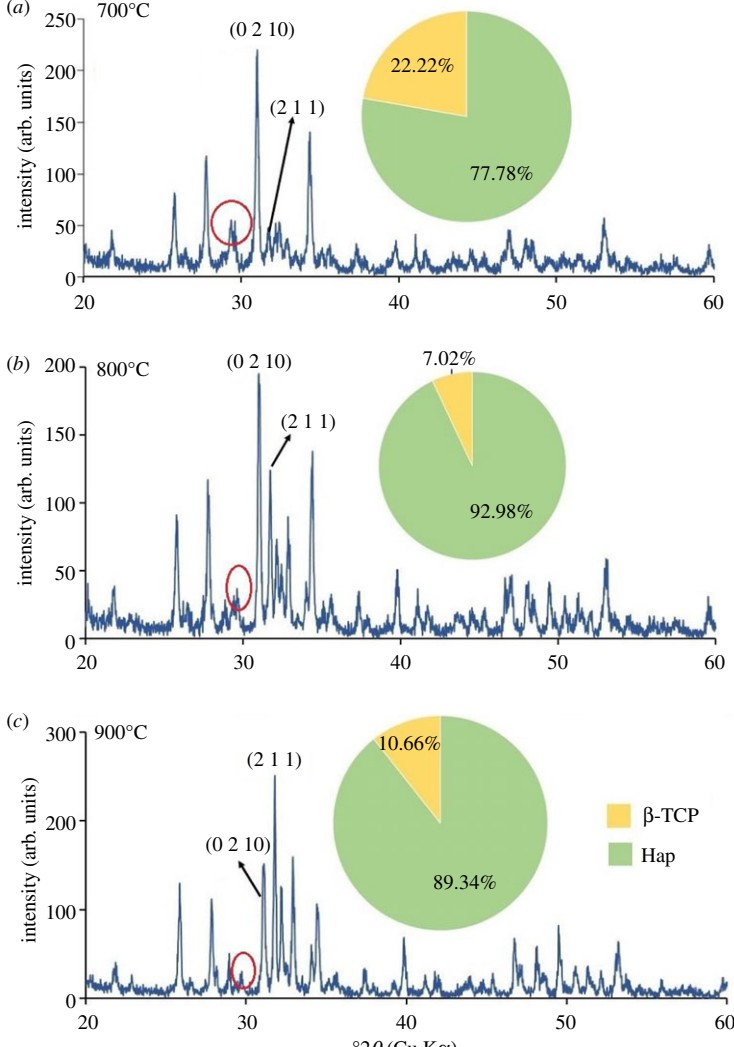

**Figure 6.** XRD pattern of Hap samples; (*a*) B-1; (*b*) B-2; (*c*) B-3.

upper range, the XRD patterns became sharper giving higher crystallinity to the products. Nevertheless, though the leading diffraction peaks were in good agreement with JCPDS cards (09-0432) representative of Hap [10,15], as an additional phase, the existence of β-tricalcium phosphate (β-TCP) was also registered. In each case, the noticeable diffraction peaks recorded at the corresponding 2θ positions were symbolic for (0 0 2), (2 1 1), (1 1 2), (3 0 0), (2 0 2), (1 3 0), (2 2 2), (2 1 3), (3 2 1) planes of Hap and (2 1 4), (0 2 10), (1 2 8), (2 2 0) planes of β-TCP [10]. It should be mentioned here that these peaks gradually transformed to more intense peaks highlighting the effect of calcination at higher temperatures (800° and 900°C). As a result of calcination at increased temperature ranges, the diffraction peaks particularly representative of Hap at (2 1 1) plane became more intense, while (1 1 2) and (3 0 0) planes appeared with almost equal intensities also supplement the formation of Hap with better crystallite form.

However, the Hap samples calcined at 700°C (sample ID: A-1, B-1, C-1, D-1, E-1, F-1) disclosed the presence of $CaCO_3$ giving a diffraction peak at 2θ position 29.39° which was conjoined with the peak of β-TCP at 29.68° indicative of (3 0 0) plane (marked with a red circle). Then again as a result of calcination at 800° and 900°C the reaction proceeds toward completion and thus the peak for $CaCO_3$ seemed to be wiped out leaving behind a single peak of β-TCP (3 0 0) plane. However, the % of carbonate content in the Hap samples was calculated from the FT-IR data as described in the following §3.2.2. Conversely, using the data of XRD patterns, the accumulated percentage of β-TCP in each Hap sample was projected from equation (3.5) [28] and shown in the pie-chart (inset of each XRD pattern) with the % of Hap. It was revealed from the pie-charts that the percentage of β-TCP in the synthesized Hap varies from 7% to 22%. Surprisingly, Hap samples (A-1, B-1, C-1, D-1, E-1, F-1)

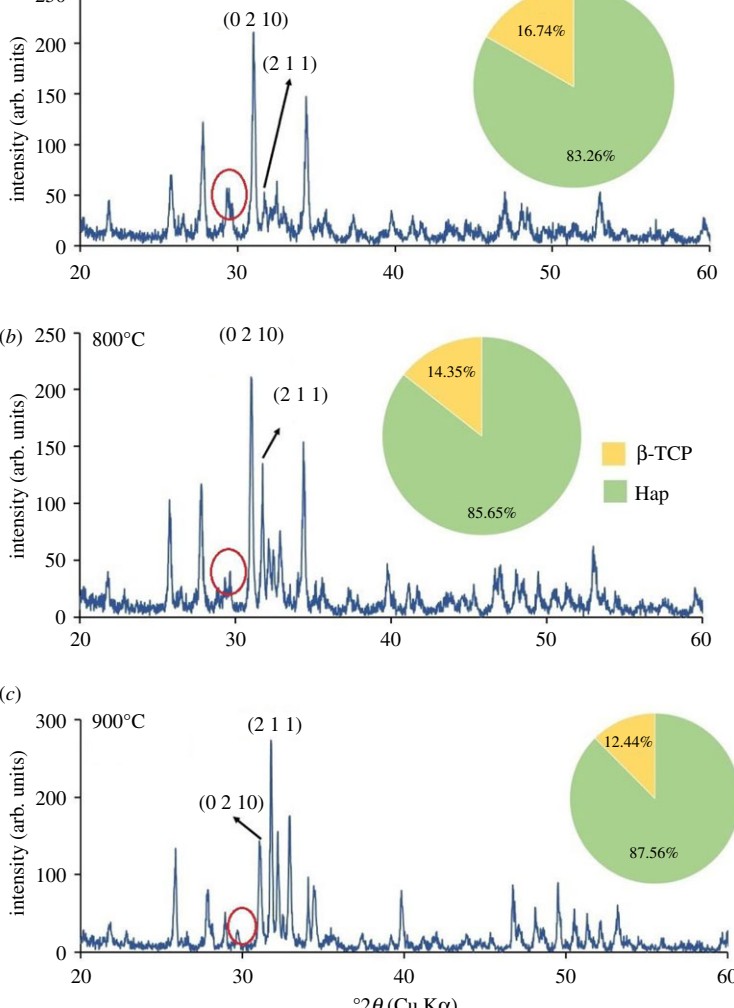

**Figure 7.** XRD pattern of Hap samples; (*a*) C-1; (*b*) C-2; (*c*) C-3.

calcined at 700°C give a significant percentage (15–22%) of β-TCP irrespective of mesh sizes. Usually, though a high-temperature calcination (1000°C) of Hap promotes the formation of β-TCP, according to Mclntosh *et al.* [29], β-tricalcium orthophosphate ($d = 2.88$) can also be formed at 700–750°C temperature and thus the formation of β-TCP at a lower temperature (i.e. at lower temperature) is fairly justified.

$$\%\beta\text{-TCP} = \frac{I_{\beta\text{-TCP(0210)}}}{I_{\text{Hap(211)}} + I_{\beta\text{-TCP(0210)}}}, \tag{3.5}$$

where $I_{\beta\text{-TCP(0210)}}$ and $I_{\text{Hap(211)}}$ signify the intensities of the characteristic peaks of β-TCP for (0 2 1 0) plane and Hap for (2 1 1) plane, respectively.

Next, to examine the effect of mesh size together with the calcination temperature on the crystallographic properties of Hap, the crystallite size ($D_c$), crystallinity degree ($Xc$) and crystallinity index (*CI*), dislocation density ($\delta$) and micro-strain ($\varepsilon$) of all the Hap samples were calculated using well-established equations [30,31] and summarized in table 1.

The data as presented in table 1 envisaged that calculated crystallite sizes are more consistent at higher calcination temperatures (i.e. 800°C and 900°C) than that obtained at 700°C. Conversely, calcination at these three selected temperatures resulted in small crystallite size values when the size of the starting materials was controlled at a lower range. However, the values of crystallinity degree ($Xc$) and crystallinity index (*CI*) carried good evidence of better Hap formation at a higher temperature. Particularly, the CI values as achieved due to the calcination at 900°C are close to that mentioned for standard Hap in JCPDS ref. code: 89-6439 (CI$_{\text{XRD}}$ for Hap is about 1.5). At 900°C

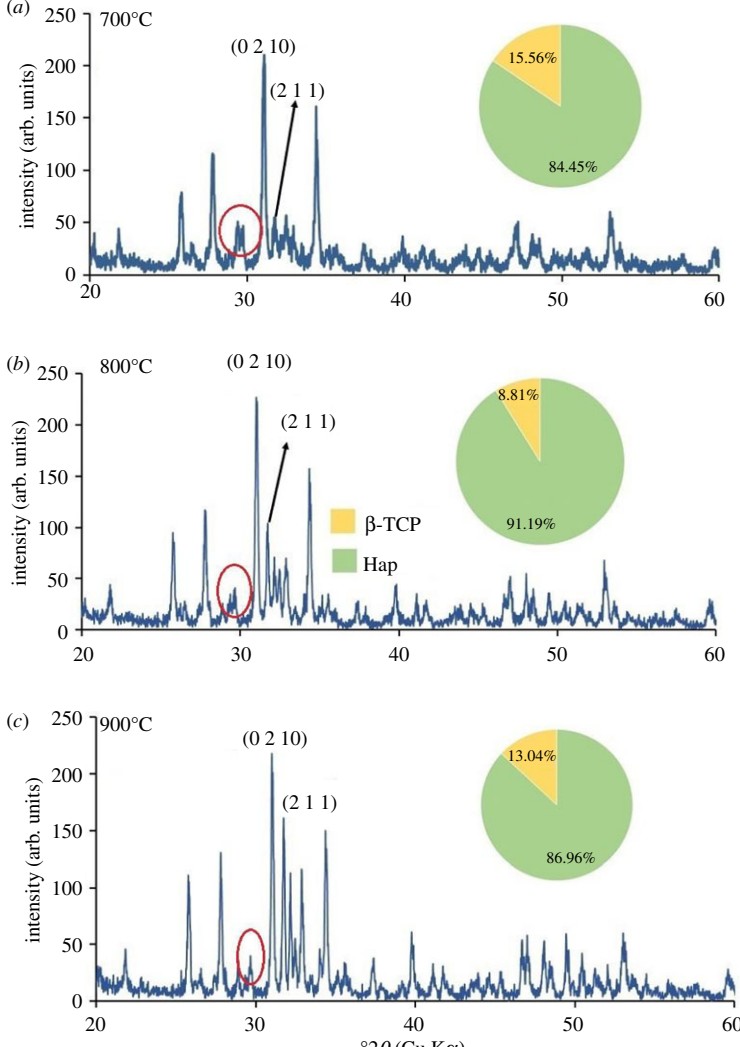

**Figure 8.** XRD pattern of Hap samples; (*a*) D-1; (*b*) D-2; (*c*) D-3.

temperature, the Hap samples (E-3 and F-3) synthesized using smaller particle sized starting materials exhibited a reduced amount of crystallinity, while this value was comparatively higher for Hap samples prepared from larger sized materials. Such observation favours the suitability of E-3 and F-3 samples for application in biomedical fields because, for biomedical application, the degree of crystallinity of Hap is significantly important as lower crystallinity increases the water solubility and protein absorption capability of Hap [32]. The calculated subsiding pattern of micro-strain and dislocation density with higher crystallite size values are also in agreement with previous results [33].

Given in figure 11 is the illustration constructed by plotting % of Hap and % of β-TCP formation as a function of sintering temperature. On the other hand, figure 12 shows the relation between the % of Hap, degree of crystallinity and the mesh sizes considered for investigation. Apparently it is visualized from table 1 and figures 11 and 12 that neither the particle size of the raw materials nor the calcination temperatures affect the crystallographic parameters (apart from crystallinity degree) data significantly. Indeed, though these values show no significant changes but in terms of percentage, the variations reflected significance. However, for practical applications of any biomaterial, the change in the values of crystallographic parameters even in nanoscale is significant. For this reason, we considered the ANOVA analysis to explore further the dual effect of particle size of the source materials and calcination temperature on the direct synthesis of Hap. Nevertheless, the ANOVA test revealed that the results of these crystallographic parameters are significantly different for the applied sintering temperatures 700°C and 800°C ($p < 0.05$), while the data obtained for calcination temperature 800°C are not significantly different from that acquired at 900°C ($p > 0.05$).

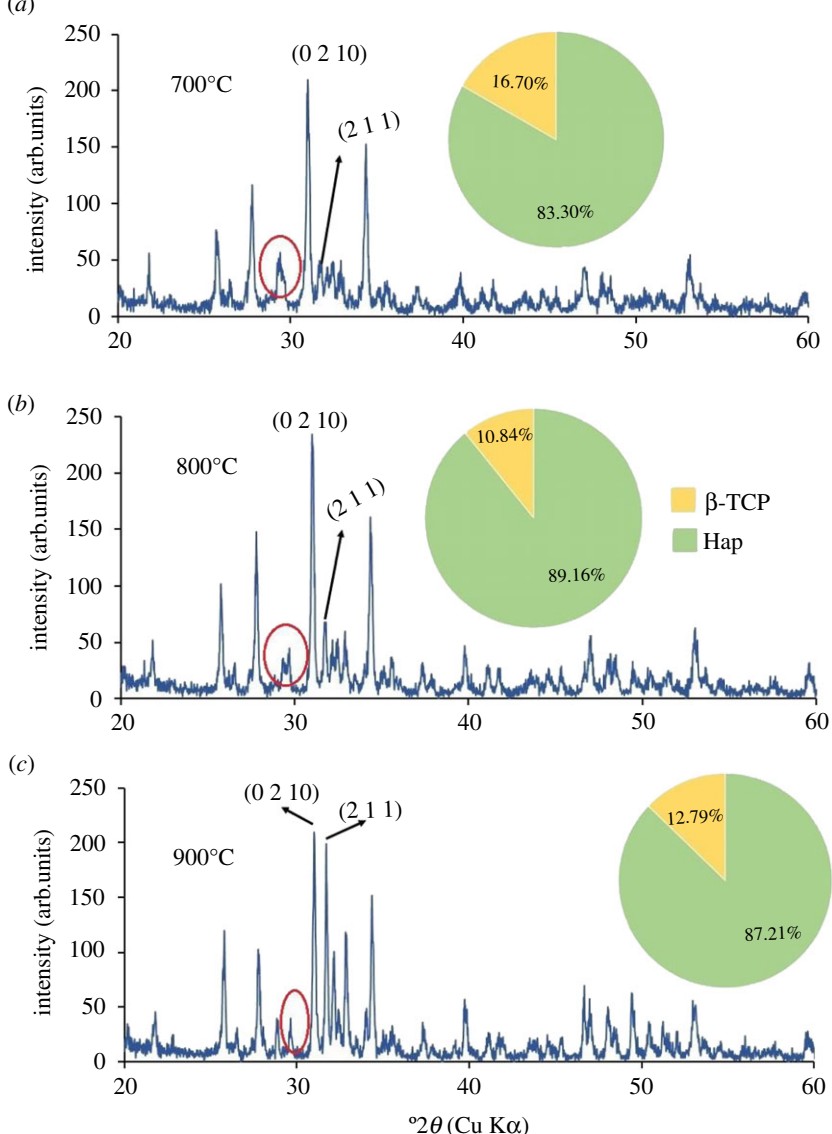

**Figure 9.** XRD pattern of Hap samples; (a) E-1; (b) E-2; (c) E-3.

### 3.2.2. Fourier transform infrared spectrophotometer and Raman analysis: molecular structure investigation

Figures 13 and 14 represent the FT-IR and Raman spectra, respectively, for a typical Hap sample. Observed data (figure 13) revealed the presence of the characteristic FT-IR band positions for $PO_4^{3-}$ group which are in well agreement with previous studies. [30] On the whole, the spectral features documented at 471, 565–603, 960–970 and 1041–1088 cm$^{-1}$ are representative of $v_2$ (symmetric bending mode of O-P-O), $v_4$ (asymmetric bending mode of O-P-O), $v_1$ (symmetric stretching mode of P-O) and $v_3$ (asymmetric stretching mode of P-O) modes of $PO_4^{3-}$ ions, respectively [30,34–36]. It is known that the latter two types of IR signals (i.e. $v_1$, symmetric and $v_3$, asymmetric) could be from P-O stretching modes of the amorphous calcium phosphate (ACP) and Hap phosphate groups [37]. But since under strict tetrahedral symmetry for ACP, $v_1$ becomes IR inactive and conversely when the symmetry in the crystal is let down from that of the free ion (e.g. in Hap), the IR signal representative of symmetric stretching appears as a weak band at 950–970 cm$^{-1}$ [36]. In this present study, the observed weak symmetric stretching band at this specific region supports the formation of Hap under the present experimental protocol. Additionally, the visible large separation between the two bands at 565 and 603 cm$^{-1}$ further ensures the existence of the crystallized apatitic phase [38,39]. The FT-IR spectrum, though, showed the presence of symmetric stretching mode of the Hap structural hydroxyl bands with low intensity in the region of 630 cm$^{-1}$ but the other band (at 3400–3600 cm$^{-1}$) for the hydroxyl stretching mode was absent. The reason for such an observation is the influence of high-temperature

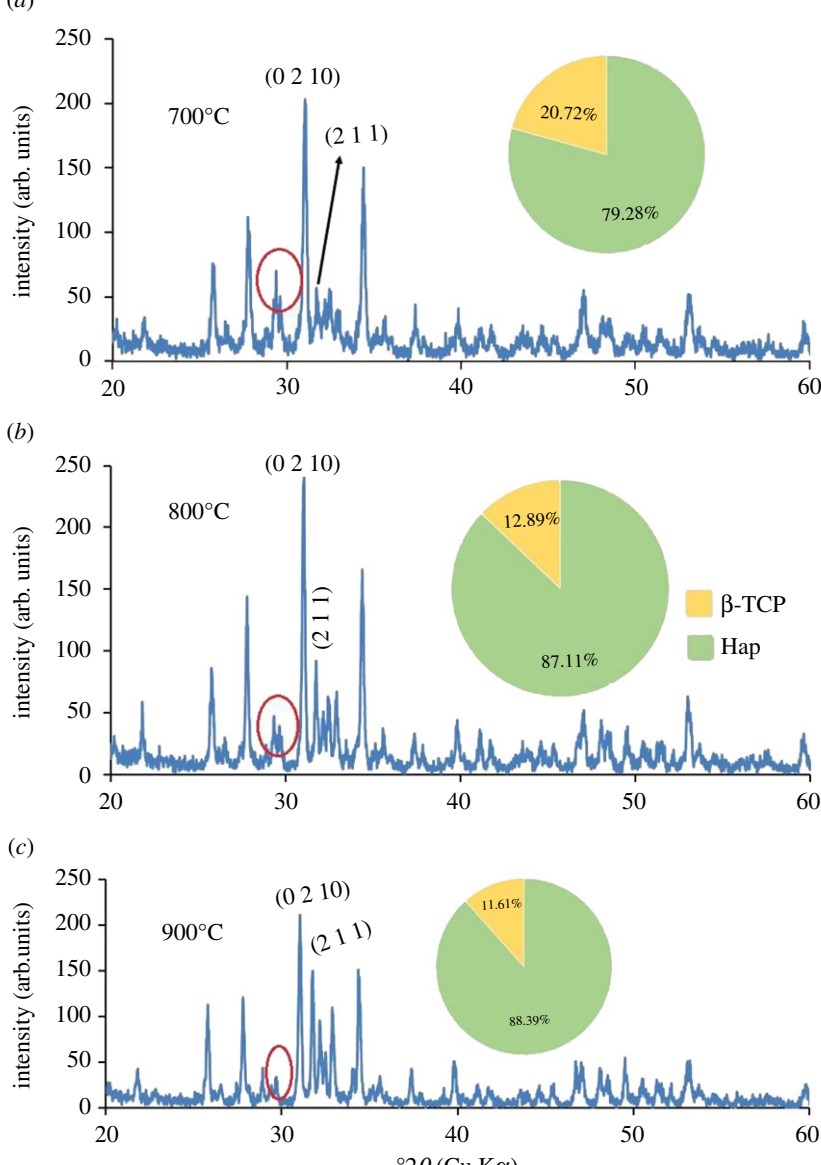

**Figure 10.** XRD pattern of Hap samples; (*a*) F-1; (*b*) F-2; (*c*) F-3.

calcination which as a result promoted the disappearance of the transmittance peak of the structural hydroxyl group as noticed in previous studies [30,35]. These intensity decreasing and disappearing characteristics of the bands at $630 \, \text{cm}^{-1}$ and $3400–3600 \, \text{cm}^{-1}$ regions, respectively, demonstrate the diminution in hydroxyl group forming β-TCP. [40]. In addition to these above-identified IR bands which confirmed the formation of Hap, few more characteristic bands of β-TCP (1160, 1072, 974 and $945 \, \text{cm}^{-1}$) [41,42] with less intensity were also noticed in the IR spectrum (figure 13). Such outcomes agree with the XRD analysis which ensured the presence of Hap and β-TCP.

The presence of the $CO_3^{2-}$ group was also evidenced in the FT-IR spectrum. The peak position of $v_3$ stretching vibration of $CO_3^{2-}$ group governs the type of substitution, i.e. A type (hydroxyl substitution) or B type (phosphate substitution) [37]. The two prominent signals as evident at 873 and $1458 \, \text{cm}^{-1}$ supported that the Hap samples are of carbonated apatite with Type B substitution which means that the $CO_3^{2-}$ substituted the $PO_4^{3-}$ in the Hap lattice [36]. However, to find out the effects of (i) particle size of the starting materials and (ii) calcination temperature on the substitution of $PO_4^{3-}$ group by $CO_3^{2-}$ group, we calculated the carbonate content in all Hap samples using the FT-IR results in equations (3.6 and 3.7) [43,44].

$$\% \text{ of } CO_3^{2-} = 13.5 \left( \frac{E_{i(1458)}}{E_{i(601)}} \right) - 0.2, \tag{3.6}$$

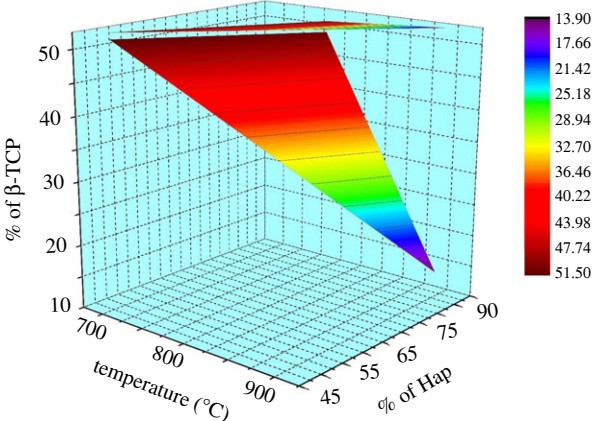

**Figure 11.** % of Hap and β-TCP as a function of sintering temperature.

**Table 1.** Calculated crystallographic parameters of Hap samples.

| ID of Hap | $D_c$ nm | CI | $X_c$ | $\delta$ | $\varepsilon$ |
|---|---|---|---|---|---|
| A-1 | 52.50 | 0.8799 | 3.545 | 0.363 | 0.1514 |
| B-1 | 83.97 | 1.2853 | 14.509 | 0.142 | 0.0887 |
| C-1 | 69.96 | 0.9758 | 8.3924 | 0.204 | 0.1064 |
| D-1 | 105.00 | 0.9641 | 28.3600 | 0.091 | 0.0708 |
| E-1 | 83.96 | 1.0139 | 14.5090 | 0.142 | 0.0887 |
| F-1 | 52.50 | 0.9737 | 3.5450 | 0.363 | 0.1416 |
| A-2 | 52.50 | 0.8825 | 3.545 | 0.363 | 0.1417 |
| B-2 | 52.49 | 1.0135 | 3.545 | 0.363 | 0.1418 |
| C-2 | 41.99 | 1.3178 | 1.8137 | 0.567 | 0.1772 |
| D-2 | 52.50 | 1.1718 | 3.545 | 0.363 | 0.1417 |
| E-2 | 59.96 | 0.9785 | 5.2831 | 0.278 | 0.1242 |
| F-2 | 59.96 | 1.0096 | 5.2831 | 0.278 | 0.1241 |
| A-3 | 70.10 | 1.4418 | 8.3924 | 0.204 | 0.1036 |
| B-3 | 68.98 | 1.2457 | 8.000 | 0.210 | 0.1054 |
| C-3 | 70.09 | 1.5102 | 8.3924 | 0.204 | 0.1037 |
| D-3 | 83.97 | 1.3485 | 14.509 | 0.142 | 0.0887 |
| E-3 | 52.50 | 1.6446 | 3.545 | 0.363 | 0.1418 |
| F-3 | 52.50 | 1.0270 | 3.545 | 0.363 | 0.1417 |

where $E_i$ is the extinction coefficient for carbonate and phosphate groups at band positions 1458 and 601 cm$^{-1}$, respectively.

$$E_i \ (1458 \text{ or } 601) = \log\left(\frac{T_2}{T_1}\right), \tag{3.7}$$

where $T_1$ and $T_2$ are the respective transmittance at the local baseline (601 and 1458 cm$^{-1}$) and the corresponding peak (601 and 1458 cm$^{-1}$).

A bar diagram representation of the % of carbonate content as obtained for each Hap sample is presented in figure 15 which shows that, for all cases, the % of carbonate content is below the maximum limit (8%) of the $CO_3^{2-}$ ion present in bone tissue Hap [43,45].

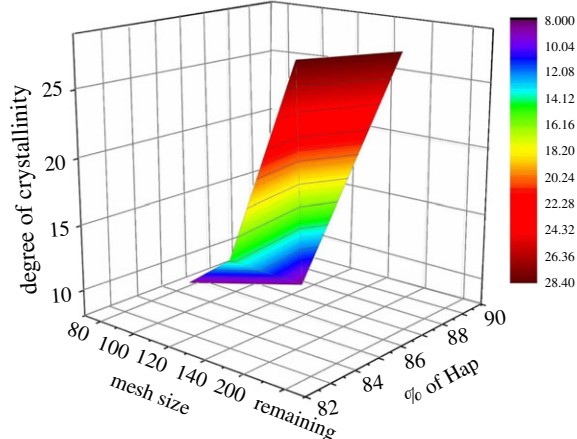

**Figure 12.** % of Hap and $X_C$ with respect to the mesh size used.

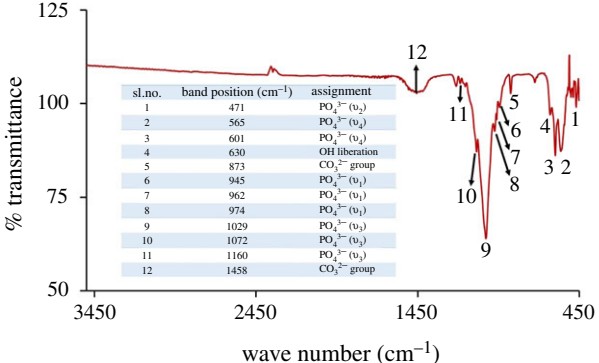

**Figure 13.** FT-IR spectrum of Hap sample.

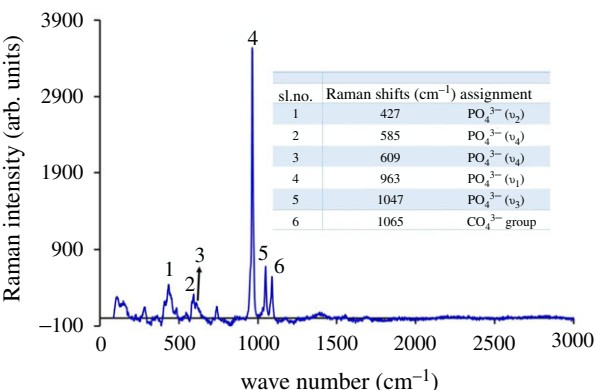

**Figure 14.** Raman spectrum of Hap sample.

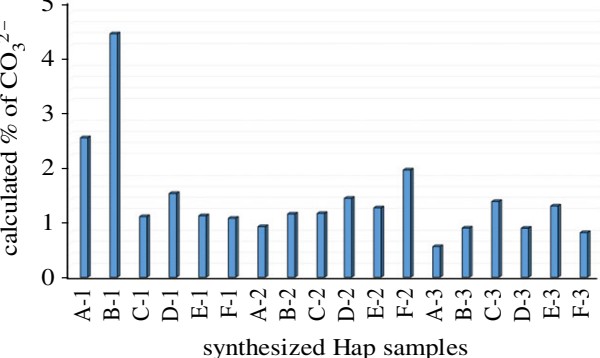

**Figure 15.** % of carbonate content in synthesized Hap samples.

Further, the existence of the $CO_3^{2-}$ group was affixed by the Raman spectroscopic data (figure 14) having the Raman shift at 1065 cm$^{-1}$ [46,47]. A common feature of symbolic Raman spectrum of B-type carbonated apatite (i.e. $CO_3^{2-}$ substituted the $PO_4^{3-}$) is the overlapping nature of symmetric stretching $v_1$ $CO_3^{2-}$ mode (at approx. 1070 cm$^{-1}$) with that of the asymmetric stretching $v_3$ $PO_4^{3-}$ band (at approx. 1050 cm$^{-1}$) [47]. But interestingly, this latter band remains visible at low $CO_3^{2-}$ content (up to 3 wt%) and then starts to be enclosed wholly by the $v_1$ $CO_3^{2-}$ peak with the significant increase in $CO_3^{2-}$ content (i.e. in bone where $CO_3^{2-}$ content is approx. 8%) [47]. Surprisingly, the present study reveals that only B-1 Hap contains about 4.5 wt% $CO_3^2$ while all other 17 Hap samples contain less than 3 wt% $CO_3^2$ (figure 15) and this justifies the presence of $v_3$ $PO_4^{3-}$ band (at 1047 cm$^{-1}$) [47]. Moreover, the Raman spectrum also documented the four distinctive vibration bands of $PO_4^{3-}$ groups which are present in Hap crystals: (i) $v_2$ bending at 427 cm$^{-1}$, (ii) $v_4$ bending at 585 and 609 cm$^{-1}$, (iii) $v_1$ stretching at 960–962 cm$^{-1}$ and (iv) $v_3$ stretching in the region 1035–1045 cm$^{-1}$. These observations are in well agreement with the previous investigation carried out by Stammeier *et al.* [48].

# 4. Conclusion

In summary, to the best of our knowledge, so far all the raw materials (either biogenic or chemicals) used to synthesize Hap were never subjected to sieve analysis. Hence, we investigated the coupled effect of single sized starting materials (80, 100, 120, 140, 200 mesh) and calcination temperature (700, 800 and 900°C) on the synthesis of Hap. Based on the ANOVA test, we conclude that the crystallographic parameters of the 18 Hap samples are significantly different at 700°C and 800°C ($p < 0.05$) which needs further investigation and we are working to explore it.

Data accessibility. The paper deals with new data which are accessible through electronic supplementary material.
Authors' contributions. M.S.H. designed the experiments, analysed the data and prepared part of the manuscript. M.M., M.B.M. and S.S. synthesized the Hap samples. M.S.I. executed TG experiments. S.A. conceived and designed the project, supervised the overall work and prepared the manuscript.
Competing interests. There is no potential conflict of interest to declare.
Funding. We received no funding for this study.
Acknowledgement. The authors are grateful to BCSIR authority for approving the R&D (ref. no. 39.02.0000.011.14.111.2019.224, Date 06.11.2019) to accomplish this research. S.S. acknowledges BCSIR for AAMS Postgraduate Fellowship. Assistance from CARF, BCSIR to acquire Raman data is also appreciated.

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
