## [Peer Review File · Royal Society Open Science]

Review History

RSOS-210684.R0 (Original submission)

Review form: Reviewer 1

Is the manuscript scientifically sound in its present form?

Yes

Are the interpretations and conclusions justified by the results?

Yes

Is the language acceptable?

Yes

Do you have any ethical concerns with this paper?

No

Have you any concerns about statistical analyses in this paper?

No

Recommendation?

Accept with minor revision (please list in comments)

Comments to the Author(s)

In this manuscript the authors described the effect of controlled particle size (obtained by using 80, 100, 120, 140 and 200 mesh) of the source materials on the synthesis of well-known biomaterial, hydroxyapatite. In addition to this, we also mapped the consequence of applied temperature (700°C, 800°C and 900°C) on the crystallographic properties and phase composition of the obtained Hap. Nevertheless, though with hydroxyapatite, in each case β -tricalcium phosphate (β -TCP) was registered as the secondary phase but the ANOVA test revealed that the results of the crystallographic parameters are significantly different for the applied sintering temperature 700°C and 800°C ($p < 0.05$) while the data obtained for calcination temperature 800°C are not significantly different from that acquired at 900°C ($P > 0.05$).

So I recommend the Manuscript to be accepted for publication. However, before publication minor revisions are required to improve the manuscript.

1. In present manuscript shows more number of self-citations. Rather, I suggest authors to include recently available appropriate references related to the current work.
2. Page No. 7 (Table no. 1) ... Next to examine the effect of mesh size together with the calcination temperature on the crystallographic properties of Hap, the crystallite size (D_c), crystallinity degree (X_c) and crystallinity index (CI), dislocation density (δ) and micro-strain (ϵ) of all the Hap samples were calculated using well-established equations and summarized. Authors should justify it by adding more discussion and references.
3. Figure 13, In the FT-IR spectrum, Give the more information of presence of Phosphate band for the formation of Hap. Authors should justify it by adding more discussion and references.
4. Figure 14, In the Raman Spectra, elaborate more with adding more discussion and references.
5. There are few type grammatical errors; authors need to check throughout the manuscript.

Review form: Reviewer 2

Is the manuscript scientifically sound in its present form?

Yes

Are the interpretations and conclusions justified by the results?

Yes

Is the language acceptable?

Yes

Do you have any ethical concerns with this paper?

No

Have you any concerns about statistical analyses in this paper?

No

Recommendation?

Accept with minor revision (please list in comments)

Comments to the Author(s)

The paper reports a significant finding about the synthesis of HAP, but certain facts has to be made clear before it being accepted for publication (see Appendix A).

Decision letter (RSOS-210684.R0)

Dear Dr AHMED:

Title: Coupled effect of particle size of the source materials and calcination temperature on the direct synthesis of hydroxyapatite
Manuscript ID: RSOS-210684

Thank you for submitting the above manuscript to Royal Society Open Science. On behalf of the Editors and the Royal Society of Chemistry, I am pleased to inform you that your manuscript will be accepted for publication in Royal Society Open Science subject to minor revision in accordance with the referee suggestions. Please find the reviewers' comments at the end of this email.

The reviewers and handling editors have recommended publication, but also suggest some minor revisions to your manuscript. Therefore, I invite you to respond to the comments and revise your manuscript.

Because the schedule for publication is very tight, it is a condition of publication that you submit the revised version of your manuscript before 15-Jul-2021. Please note that the revision deadline will expire at 00.00am on this date. If you do not think you will be able to meet this date please let me know immediately.

- 1) A text file of the manuscript (tex, txt, rtf, docx or doc), references, tables (including captions) and figure captions. Do not upload a PDF as your "Main Document".
- 2) A separate electronic file of each figure (EPS or print-quality PDF preferred (either format should be produced directly from original creation package), or original software format)

- 3) Included a 100 word media summary of your paper when requested at submission. Please ensure you have entered correct contact details (email, institution and telephone) in your user account
- 4) Included the raw data to support the claims made in your paper. You can either include your data as electronic supplementary material or upload to a repository and include the relevant doi within your manuscript
- 5) All supplementary materials accompanying an accepted article will be treated as in their final form. Note that the Royal Society will neither edit nor typeset supplementary material and it will be hosted as provided. Please ensure that the supplementary material includes the paper details where possible (authors, article title, journal name).

Kind regards,
Dr Laura Smith
Publishing Editor, Journals

On behalf of the Subject Editor Professor Anthony Stace and the Associate Editor Dr Dattatray Late.

RSC Associate Editor:
Comments to the Author:
Accept with minor revisions

RSC Subject Editor:
Comments to the Author:
(There are no comments.)

Reviewer comments to Author:

Reviewer: 1

Comments to the Author(s)

In this manuscript the authors described the effect of controlled particle size (obtained by using 80, 100, 120, 140 and 200 mesh) of the source materials on the synthesis of well-known

biomaterial, hydroxyapatite. In addition to this, we also mapped the consequence of applied temperature (700°C, 800°C and 900°C) on the crystallographic properties and phase composition of the obtained Hap. Nevertheless, though with hydroxyapatite, in each case β -tricalcium phosphate (β -TCP) was registered as the secondary phase but the ANOVA test revealed that the results of the crystallographic parameters are significantly different for the applied sintering temperature 700°C and 800°C ($p < 0.05$) while the data obtained for calcination temperature 800°C are not significantly different from that acquired at 900°C ($P > 0.05$).

So I recommend the Manuscript to be accepted for publication. However, before publication minor revisions are required to improve the manuscript.

1. In present manuscript shows more number of self-citations. Rather, I suggest authors to include recently available appropriate references related to the current work.
2. Page No. 7 (Table no. 1) ... Next to examine the effect of mesh size together with the calcination temperature on the crystallographic properties of Hap, the crystallite size (Dc), crystallinity degree (Xc) and crystallinity index (CI), dislocation density (δ) and micro-strain (ϵ) of all the Hap samples were calculated using well-established equations and summarized. Authors should justify it by adding more discussion and references.
3. Figure 13, In the FT-IR spectrum, Give the more information of presence of Phosphate band for the formation of Hap. Authors should justify it by adding more discussion and references.
4. Figure 14, In the Raman Spectra, elaborate more with adding more discussion and references.
5. There are few type grammatical errors; authors need to check throughout the manuscript.

Reviewer: 2

Comments to the Author(s)

The paper reports a significant finding about the synthesis of HAP, but certain facts has to be made clear before it being accepted for publication. Please see attached file

Author's Response to Decision Letter for (RSOS-210684.R0)

See Appendix B.

RSOS-210684.R1 (Revision)

Review form: Reviewer 1

Is the manuscript scientifically sound in its present form?

Yes

Are the interpretations and conclusions justified by the results?

Yes

Is the language acceptable?

Yes

Do you have any ethical concerns with this paper?

No

Have you any concerns about statistical analyses in this paper?

No

Recommendation?

Accept as is

Comments to the Author(s)

Arthurs have satisfactory gave the answers to comments in the review manuscript

Review form: Reviewer 3 (Balakumar Subramanian)

Is the manuscript scientifically sound in its present form?

Yes

Are the interpretations and conclusions justified by the results?

Yes

Is the language acceptable?

Yes

Do you have any ethical concerns with this paper?

No

Have you any concerns about statistical analyses in this paper?

No

Recommendation?

Accept as is

Comments to the Author(s)

As they revised the manuscript satisfactorily, it may be accepted in the present form

Decision letter (RSOS-210684.R1)

Dear Dr AHMED:

Title: Coupled effect of particle size of the source materials and calcination temperature on the direct synthesis of hydroxyapatite
Manuscript ID: RSOS-210684.R1

It is a pleasure to accept your manuscript in its current form for publication in Royal Society Open Science. The chemistry content of Royal Society Open Science is published in collaboration with the Royal Society of Chemistry.

Yours sincerely,
Dr Ellis Wilde
Publishing Editor, Journals

On behalf of the Subject Editor Professor Anthony Stace and the Associate Editor Dr Dattatray Late.

RSC Subject Editor
Comments to the Author:
Accept as is

RSC Associate Editor
Comments to the Author:
(There are no comments.)

Reviewer(s)' Comments to Author:
Reviewer: 3
Comments to the Author(s)
As they revised the manuscript satisfactorily, it may be accepted in the present form

Reviewer: 1
Comments to the Author(s)
Arthurs have satisfactory gave the answers to comments in the review manuscript

Appendix A

The manuscript entitled “**Coupled effect of particle size of the source materials and calcination temperature on the direct synthesis of hydroxyapatite**” submitted to *Royal Society Open Science* appears to be an interesting piece of scientific finding done in the area of biomaterials science. The findings are well supported by the experimental results followed by adequate discussions to support the findings. The manuscript could be published to a wider audience provided the authors could make some corrections regarding few details they have provided in the manuscript.

Comments

- #1 In the introduction, the authors have stated that the paper is “intended to examine the coupled effect of particle size of the source materials and calcination temperature to synthesize hydroxyapatite by solid state method, but later they state “Apparently it is visualized from Table 1 and Figs. 11 & 12 that neither the particle size of the raw materials nor the calcination temperatures effect the crystallographic parameters (apart from crystallinity degree) data significantly”. Then what do they really intend to report in this paper? The authors need to clarify this.
- #2 For synthesizing the HAP samples, heating was achieved through oven and calcination. The temperature for oven drying and the time duration was not mentioned. It would give more clarity.
- #3 While reporting for HAP through FTIR (**Fig. 13**), a typical band around 3450 cm^{-1} which is characteristic of HAP otherwise it could be other calcium phosphate also. Why has it not been observed or accounted?
- #4 The XRD analysis of samples (Figs. 5 -10) show significant percentage of β - TCP even after 700°C , which should have been otherwise showing a complete phase transformation to HAP. Can the authors give their rationale?
- #5 Significant amount of CO_2 is present in the resulting HAP samples raises doubt whether it was incomplete decomposition of calcite or just atmospheric trapping?
- #6 There are few grammatical errors which could be corrected during the revision of the manuscript

Overall Comment : The manuscript could be accepted after “**Minor Revision**”

Appendix B

Royal Society Open Science manuscript ID: RSOS-210684

Title: Coupled effect of particle size of the source materials and calcination temperature on the direct synthesis of hydroxyapatite

Answer to Reviewers' Reports

Reviewer: 1

1. In present manuscript shows more number of self-citations. Rather, I suggest authors to include recently available appropriate references related to the current work.

Ans. Following references have been incorporated replacing the self-citation references 7, 17 and 20 accordingly and ref. 20 is now re-numbered as **21**.

New. Ref. 7 A. Mocanu, O. Cadar, P. T. Frangopol, I. Petean, G. Tomoaia, G-A. Paltinean, C. P. Racz, O. Horovitz and M. Tomoaia-Cotisel, *R. Soc. Open Sci.* 2020, 8, 201785.

New. Ref. 17 S. Hokkanen, B. Doshi, V. Srivastava, L. Puro and R. Koivula, *Environmental progress & sustainable energy*, 2019, 38(5). <https://doi.org/10.1002/ep.13147>.

New. Ref. 21 S. Senthilkumar, V. Dhivya, M. Sathya and A. Rajendran, *Journal of Experimental Nanoscience*, 2021, 16:1, 160-180, DOI: 10.1080/17458080.2021.1931685

2. Page No. 7 (Table no. 1) ... Next to examine the effect of mesh size together with the calcination temperature on the crystallographic properties of Hap, the crystallite size (Dc), crystallinity degree (Xc) and crystallinity index (CI), dislocation density (δ) and micro-strain (ϵ) of all the Hap samples were calculated using well-established equations and summarized. Authors should justify it by adding more discussion and references.

Ans. Justified accordingly by adding more discussion and references (In the manuscript justification is added after Table 1 (marked with red ink) and the newly incorporated refs. are 23 & 24.

3. Figure 13, In the FT-IR spectrum, Give the more information of presence of Phosphate band for the formation of Hap. Authors should justify it by adding more discussion and references.

Ans. Justified accordingly by adding more discussion and references (In the manuscript justification is added in Section 3.2.2 (marked with red ink).

4. Figure 14, In the Raman Spectra, elaborate more with adding more discussion and references.

Ans. Justified accordingly by adding more discussion and references (In the manuscript justification is added in Section 3.2.2 (marked with red ink).

5. There are few type grammatical errors; authors need to check throughout the manuscript.

Ans. Checked and corrected (marked with red ink in the text)

Reviewer: 2

1. In the introduction, the authors have stated that the paper is “intended to examine the coupled effect of particle size of the source materials and calcination temperature to synthesize hydroxyapatite by solid state method, but later they state “Apparently it is visualized from Table 1 and Figs. 11 & 12 that neither the particle size of the raw materials nor the calcination temperatures effect the crystallographic parameters (apart from crystallinity degree) data significantly”. Then what do they really intend to report in this paper? The authors need to clarify this.

Ans. We, here in this paper attempted to examine the dual effect of particle size of the source materials and calcination temperature on the direct synthesis of hydroxyapatite. Consequently, we investigated various crystallographic properties e.g. crystallite size (D_c), crystallinity degree (X_c) and crystallinity index (CI), dislocation density (δ) and micro-strain (ϵ) of all the Hap samples prepared under the present experimental protocol. The data as shown in Table 1 and Figs. 11 & 12 though apparently shows no significant changes but in terms of percentage the changes were important. For instance, the calculated crystallite size (D_c) for C-2 (120 mesh sieved and calcined at 800°C) Hap is 41 nm while for D-1 (140 mesh sieved and calcined at 700°C) it is 105 nm. Hence an inflation of 256% was achieved which is obviously significant. Again crystallinity index (CI) varied from 0.8 to 1.6 (i.e. 200% variation), dislocation density (δ) varied from 0.09 to 0.56 (622% variation) etc. Though the values seem to be small but their variations are large and significant. For this reason, we considered the ANOVA analysis to make a conclusion regarding the dual effect of particle size of the source materials and calcination temperature on the direct synthesis of Hap. Necessary clarification added in the relevant section (marked with red ink)

2. For synthesizing the HAP samples, heating was achieved through oven and calcination. The temperature for oven drying and the time duration was not mentioned. It would give more clarity.

Ans: The temperature for oven drying and the time duration has been mentioned accordingly in section 3.1 (1st paragraph, 2nd & 3rd lines, marked with red ink)

3. While reporting for HAP through FTIR (Fig. 13), a typical band around 3450 cm⁻¹ which is characteristic of HAP otherwise it could be other calcium phosphate also. Why has it not been observed or accounted?

Ans: Necessary explanation has been given in section 3.2.2 (text marked with red ink)

4. The XRD analysis of samples (Figs. 5 -10) show significant percentage of β - TCP even after 700 °C, which should have been otherwise showing a complete phase transformation to HAP. Can the authors give their rationale?

Ans. Justified accordingly by adding more discussion and references (In the manuscript justification is added in Section 3.2.2 (marked with red ink).

5. Significant amount of CO₂ is present in the resulting HAP samples raises doubt whether it was incomplete decomposition of calcite or just atmospheric trapping?

Ans. Indeed, the synthesized 18 Hap samples contains CO₃²⁻ (not CO₂) and supports the formation of carbonated apatite commonly known as B-type substituted apatite (where the CO₃²⁻ substitutes the PO₄³⁻ in the Hap lattice). Our XRD and FT-IR investigations also support the formation of this apatite. However, the amount of CO₃²⁻ present in these samples varies from 0.6 – 4.5% which are within the acceptable limit, as natural bone contains ~ 8% CO₃²⁻. The possible source of this CO₃²⁻ in the Hap samples could be either the calcination of calcite or atmospheric trapping or both.

6. There are few grammatical errors which could be corrected during the revision of the manuscript.

Ans. Grammatical errors are corrected (marked with red ink in the text)